# Biological Age in Relation to Somatic, Physiological, and Swimming Kinematic Indices as Predictors of 100 m Front Crawl Performance in Young Female Swimmers

**DOI:** 10.3390/ijerph18116062

**Published:** 2021-06-04

**Authors:** Kamil Sokołowski, Marek Strzała, Arkadiusz Stanula, Łukasz Kryst, Artur Radecki-Pawlik, Piotr Krężałek, Thomas Rosemann, Beat Knechtle

**Affiliations:** 1Department of Water Sports, Faculty of Physical Education and Sport, University of Physical Education, 31-541 Kraków, Poland; sokolowski.kc@gmail.com (K.S.); marek.strzala@awf.krakow.pl (M.S.); 2Institute of Sport Sciences, Jerzy Kukuczka Academy of Physical Education, 40-065 Katowice, Poland; 3Department of Anthropology, Faculty of Physical Education, University of Physical Education, 31-541 Kraków, Poland; lukasz.kryst@awf.krakow.pl; 4Institute of Structural Mechanics, Faculty of Civil Engineering, Cracow University of Technology, 31-155 Kraków, Poland; rmradeck@cyf-kr.edu.pl; 5Department of Physiotherapy, Faculty of Motor Rehabilitation, University of Physical Education, 31-541 Kraków, Poland; piotr.krezalek@awf.krakow.pl; 6Institute of Primary Care, University of Zurich, 8091 Zurich, Switzerland; thomas.rosemann@usz.ch; 7Medbase St. Gallen Am Vadianplatz, 9000 St. Gallen, Switzerland

**Keywords:** female adolescents, biological maturation, swimming flume, ventilatory thresholds, front crawl swimming

## Abstract

Background: Some swimmers reach high performance level at a relatively young age. The purpose of this study is to determine the relationship between adolescent female swimmers’ 100 m front crawl race ($V_{total100}$) and several anthropometry, body composition, and physiological and specific strength indices. Methods: Nineteen adolescent female swimmers were examined for biological age (*BA*) and body composition. Oxygen uptake was measured during water-flume stage-test front crawl swimming with ventilatory thresholds examination. Specific strength indices were assessed during 30 s of tethered swimming. Stroke rate (*SR*), stroke length (*SL*), and stroke index (*SI*) were also examined. Results: *BA* was strongly correlated with anthropometrics and tethered swimming strength indices, and showed moderate to strong correlation with ventilatory thresholds. Speed of swimming in the race was moderately to largely correlated with speed at $\dot{V}O_{2\;}max$−$V_{\dot{V}O_{2}max}$ (r = 0.47–0.55; *p* < 0.05)—ventilatory thresholds ($V_{AT}$,$\;V_{RCP}$) (r = 0.50–0.85; *p* < 0.05), *SL* (r = 0.58–0.62; *p* < 0.05), and *SI* (r = 0.79–0.81; *p* < 0.01). Conclusion: Results confirmed a significant role of biological maturation mediation on body composition and body size, ventilatory indices, and specific strength indices. *BA* was not a significant mediation factor influencing the swimming kinematics (*SL, SI*) and speeds of $V_{AT}$, $V_{RCP}$ or $V_{\dot{V}O_{2\;}max}$, which were strong predictors of the 100 m race.

## 1. Introduction

Female competitive swimming is a sport in which systematic training sessions and high training loads are implemented at an early age [1,2]. Training more and more extensively with gradually and periodically increasing intensity is undertaken along with the phenomenon of progressive biological maturation. The onset of puberty and mental, morphological, and physiological maturation interacts with the development of determinant factors, which affect swimming performance at the age-group level [3,4]. In considering competitive achievements, individual variations in intervals between earlier or later occurrences of growth spurts in one competitive age group cause variations in motor abilities as well as performance. Young swimmers at higher maturity levels are more likely to perform better than their less mature peers due to greater aerobic and anaerobic abilities [5,6,7,8]. Indeed, physical parameters such as body height, free fat mass or more athletic constitution, or bioenergetical indices such as swimming oxygen consumption have been shown in previous studies [2,9] to have an influence on the swimming performance of young women swimmers. Nevertheless, considering the swimming elite in the long term, Timakova and Klyuchnikova [9] pointed out that female swimmers with relatively slower maturation prevailed over the others, reaching high performance levels in adulthood. Monitoring growth as well as somatic and physiological traits in relation to biological maturation is therefore crucial to young athletes’ training optimization and expectations of adequate sports performance [8]. Anthropometric measurements from earlier studies [10] revealed a strong relationship between morphological indices and swimming performance. Cochrane et al. [11] stated that morphological characteristics of young swimmers influence swimming performance and vary by events. Vorontsov et al. [7] claim that the strongest effect of maturity on physical conditioning and strength was indicated in the group of girls aged 13–14.

The number of studies of young swimmers in which oxygen uptake is measured directly in swimming is still very limited [12]. Malina et al. [13] noted that aerobic power of more mature female and male adolescent trained children was higher than in their less mature peers (differences ranging from 0.2 up to 1.0 L·$\min^{-1}$). Anaerobic metabolism is more developed in adults than in children, and it indicates that the maturation process influences the level of anaerobic energy production [4,14]. Tethered swimming is the most specific in-water test for strength measurement [15]. Nevertheless, according to Moran et al. [6], anaerobic power, peak force of arms, and free fat mass values are mediated by maturity, and swimmers who are already at peak high velocity are more likely to respond strongly to strength training.

When training age-group young women swimmers, technique should be the main concern, because it is one of the most important factors influencing present and future performance. Stroke length and stroke rate or stroke index are essential for an efficient swimming technique. Therefore, here the authors aim to analyze in adolescent female swimmers the influence of a set of morphological, specific strength, and physiological indices on the 100 m front crawl swimming race.

These studies are conducted while considering the impact (correlation or mediation effect) of biological age (*BA*) on swimming determinant factors: (a) body composition, (b) physiological and specific strength, and (c) kinematic indices of 100 m front crawl swimming. The subsequent aim of this study is to identify a set of variables which influence 100 m front crawl performance in female swimmers, but which are not directly related to *BA*.

The authors expect that *BA* will differentially influence the particular set of indices which could be higher for indicators related to body dimensions and strength, and lower for those related to specific swimming abilities: stroke kinematics and speed on metabolic thresholds ($V_{AT}$, $V_{RCP}$, $V_{\dot{V}O_{2\;}max}$).

## 2. Materials and Methods

### 2.1. Participants

Nineteen female swimmers (age 13.4 ± 0.26, min: 12.71, max: 13.73 years; height 1.66 ± 0.07 m; body mass 55.5 ± 9.3 kg) participated in this study. They were recruited from the most successful swimmers in their age category from the Cracow, Poland region. All of them were healthy and had licenses from the Polish Swimming Federation. All swimmers went through 4–5 years of systematic swimming, trained in at least 10 training units weekly and took part in national level competitions and national swimming championships for their age group. Despite swimming style specialization, all the participants performed in freestyle events regularly. The study was approved by the Regional Medical Chamber in Cracow on 5 June 2020 (No. 94/KBL/OIL/2020). All participants and their parents provided informed consent for their participation in intensive physical effort during this study (parents of all participants became acquainted with the study program and with a short description of the tests).

### 2.2. Body Composition and Biological Age

The body composition analyzer Tanita BC-418 (Tokyo, Japan) was used to assess segmental body composition. In addition to body mass (*BM*, kg) measurement, this device uses bioelectrical impedance analysis (BIA), a method of analyzing electrical responses to a weak electrical current introduced into the body. It is a research method that allows one to assess the human body composition with regard to extracellular and intracellular water, fat and lean mass, and cell mass, based on the differentiation of tissue resistance [16]. The body fat estimated by BIA has almost perfect reproducibility, making it an applicable research tool in studies that investigate body composition changes. FFM estimated by BIA correlates almost perfectly with reference methods, regardless of sex. Moreover, regarding quality, BIA has shown high reproducibility (correlation coefficient between 0.95 and 0.99 [17]. BIA is a reliable method of assessing the tissue composition of the body; its reliability and validity have been recognized in many independent studies: Jackson et al. [18]; Aandstad et al. [19]; Dave et al. [20]; Cortesi et al. [21]; Vasold et al. [22]. This method is successfully used by both untrained people and by athletes of all disciplines [23,24]. The participants dressed in underwear, stood with electrodes on their bare feet, and gripped handheld electrodes. This procedure provided data on fat free mass (*FFM*, kg), total body water (*TBW*, kg), and predicted muscle mass of body segments: arms (*m_m arms_*, kg), trunk (*m_m trunk_*, kg), and legs (*m_m legs_*, kg). *FFM* and *TBW* values were also converted to percentages of *BM*. Biological age (*BA*) examinations of participants were conducted by an experienced anthropologist, who used the following calculation: *BA* = (*BH_age_* + *BM_age_*)/2, where *BH_age_* (height age) = age obtained from percentile charts (growth charts by The Children’s Memorial Health Institute; 50th percentile was used to align height and mass with age) on the basis of the participant’s body height and *BM_age_* (mass age) = age obtained from percentile charts on the basis of the participant’s body mass (growth charts by The Children’s Memorial Health Institute, standardized and validated for the Polish population; 50th percentile was used to align height and mass with age). Additionally, pubertal development was assessed, by an experienced, formally trained anthropologist. Namely, Tanner stages based on pubic hair scale were estimated [25] and the date of menarche was obtained retrospectively (year, month, and, if possible, exact day). The recall method has previously been widely used in research, as well as justified as a reliable way to obtain the age of menarche [26,27]. It is a validated method, used for several dozen years all over the world [28,29]. It was also used many times in the youth population from Cracow [30].

Participants took part in two test trials. One contained tethered swimming, 100 m front crawl race and anthropometric measurements. During the second one, separated by 48 h, stage test in water flume was implemented. Before each test, the swimmers completed a 1000 m in-water warm up with low to moderate intensity.

### 2.3. Stage Test

The stage in a water flume (Figure 1) was conducted in a laboratory-controlled environment. All swimmers were informed about the testing procedure and performed a 1000 m in-water warm up, as before a competition. Participants wore a nose clip and were attached to a respiratory valve system with an expired air analyzer (Start 2000 MES, Poland). One minute of slow-paced swimming ensured their adjustment to the testing conditions. After an initial speed of 0.93 m s^−^^1^ providing moderate intensity of$\;30–40\%\;{\dot{V}O}_{2}max$, every two minutes a whistle signaled for them to increase speed by 0.06 m s^−1^. Breath by breath, exhaled air was continuously sampled and saved (Ergo2000M software MES, Poland). $\dot{V}O_{2}max$, aerobic threshold (*AT*), and respiratory compensation point (*RCP*) were estimated [31]. The test was terminated after complete exhaustion and inability to maintain required swimming pace, reaching criteria of $\dot{V}O_{2}max$ examination [32]. Speed of the water flow and oxygen uptake (${VO}_{2}AT$ and $V_{AT}$, $VO_{2}RCP$ and $V_{RCP}$, $\dot{V}O_{2}max$ and $V_{\dot{V}O_{2}max}$) were assessed. In our study, all participants met the mentioned criteria, with RER (1.18 ± 0.17).

### 2.4. Tethered Swimming Test

In the 30 s tethered swimming test, participants wore a waist belt and were connected to a steel pole (fixing point 0.49 m above the surface) by a 5.65 m steel cable and attached dynamometer (with 100 Hz recording frequency). The following indices were collected:
Maximum value of force ($F_{max}$, N);Average value of force ($F_{ave}$, N);Force decline ($F_{decline}$, N), calculated from decrease in average force production in 0–10 and 20–30 s of the recording duration;Average impulse per single cycle ($I_{ave}$, N·s), defined as the integral of force over a period of time containing all full cycles divided by a number of completed cycles:
$$I_{ave}=\frac{\int_{t_{0}}^{t_{1}}Fdt}{n}$$
where: *t*_0_ is the beginning of the first full cycle and *t*_1_ is the ending of the last full cycle in the 30 s period.

### 2.5. Swimming Race

The 100 m race was carried out in a 25 m swimming pool that meets International Swimming Federation (FINA) requirements. The final results and split times of the trials were measured with an automatic timing device (Omega, Switzerland). Each one of the trial series was performed by three to four swimmers in order to imitate competition conditions. All trials were recorded with a (JVC GC-PX100BE, Japan) camera with 50 Hz framing. The camera was placed on a tripod at the bleachers, seven meters above the water surface on an extension in the middle point of the pool. The swimmers started from the blocks at the sound signal. Markers were placed at the side of the pool to indicate the line of 7 m from each of the walls and 10 m from the starting block. The pool was divided into three zones: (a) I turn zone (7 m), (b) surface swimming zone (11 m), and (c) II turn zone (7 m). Including the first 10 m start zone, it resulted in a) 59 m for $V_{STF}$ (start, turn, finish velocity) calculation and b) 41 m for $V_{surface}$ (surface swimming velocity) examination. Times for separate sectors were measured when swimmers’ heads crossed the imaginary line linking markers at the sides of the pool (Kinovea ver. 0.8.15 software). The 100 m front crawl speed ($V_{total100}$) was calculated from the final time taken to complete the 100 m distance. The average *SR* (cycle min^−^^1^) (ICC = 0.99, 95%, CI = 0.960–0.997) was calculated from 12 cycles (3 cycles from each of the 4 laps, measured in the surface swimming zone). *SL* was calculated from the 11 m surface swimming zones, during 4 laps. Stroke index *SI* ($m^{2}$·cycle s ^−1^) was calculated as: $SI=V_{surface}\cdot SL$.

### 2.6. Statistical Analysis

Individual, mean, and standard deviations (SD) computations for descriptive analysis were obtained for all studied variables. Measures of skewness, kurtosis and the Shapiro–Wilk test were used to assess the normality and homogeneity of the data.

One-way ANOVA with repeated measures and Tukey’s HSD post hoc test were carried out to detect and present differences between: ($V_{total100}$, $V_{surface}$, $V_{STF}$). To identify the relationship between the variables, the Pearson correlations were computed between:
(a)Anthropometric, body composition indices and all the indices, tethered swimming test ($F_{max\;}$, $F_{ave}$, $F_{decline}$, $I_{ave}$) and swimming speed ($V_{total100}$, $V_{surface}$,$\;V_{STF}$),(b)Stage test and swimming speeds or tethered swimming variables, and(c)*SR*, *SL*, *SI* and $V_{total100}$, $V_{surface}$, $V_{STF}$.

To examine the possible mediation effect of *BA* on variables of $V_{RCP}$, $V_{\dot{V}O_{2}max}$, $V_{AT}$, *SI* and *SL*, which correlated the most with $V_{total100}$, were tested by mediation analysis with the Sobel test. Mediation analysis was made on the basis of three regression models [33]. The tests were conducted with STATISTICA 13.1 software (TIBCO Software Inc, Palo Alto, CA, USA). A significance level of *p* ≤ 0.05 was established. Mediation analysis was prepared using R software ver. 4.5.0 with *mediation* package.

The magnitude of the correlations was determined using the modified scale by Hopkins (Hopkins, WG. Measures of reliability in sports medicine and science. Sports Med 30: 1–15, 2000.): trivial: r < 0.1; low: 0.1–0.3; moderate: 0.3–0.5; high: 0.5–0.7; very high: 0.7–0.9; nearly perfect > 0.9; and perfect: 1. To get significant results (*p* < 0.05) with sufficient power (80%) to detect at least a correlation coefficient of 0.6, the minimum required sample size for this study is 19. The formula for calculation is based on two-tailed test (Guenther, W C. 1977 Desk Calculation of Probabilities for the Distribution of the Sample Correlation Coefficient. The American Statistician).

## 3. Results

There was a significant difference between measured average speed values of: $V_{total100}$, $V_{surface}$, and $V_{STF}$ (F = 127.0, *p* ≤ 0.001). Post hoc Tukey’s (HSD) test confirmed significant differences among all of the measured averages (*p* ≤ 0.001). Figure 2 presents differences between all of the three analyzed averages.

Biological age (*BA*) presents high correlation relationship with body composition indices of *FFM* and *TBW* (r = −0.56, *p* ≤ 0.05). The highest correlations (r = 0.88 to 0.92, *p* ≤ 0.001) were found between biological age and *h*, BM, *FFM*, *TBW*, and $m_{m\;total}$ (Table 1).

The anthropometric indices of height, body mass, and muscle mass of particular body parts all showed significant moderate to very high relationship with tethered swimming indices. All tethered swimming indices correlated moderately to very highly with $m_{m\;total}$, $m_{m\;arms}$ (3.43 ± 0.54),$\;m_{m\;trunk}\;($24.03 ± 3.14), and$\;m_{m\;legs}\;($6.43 ± 0.93). *BA* was correlated at a very high level with maximal propulsion force and average impulse (Table 2).

There was no significant correlation between body composition, tethered swimming indices, and free swimming speeds: $V_{total100}$, $V_{surface}$, $V_{STF.}$

The anthropometric, body composition indices correlated moderately to very highly with: $\dot{V}O_{2}AT$, $\dot{V}O_{2}RCP$, and $\dot{V}O_{2}max$ values. Significant correlations were observed between the tethered and stage tests $F_{max}$ and ventilatory indices $\dot{V}O_{2}AT$ and $\dot{V}O_{2}RCP$ (r = 0.53 and r = 0.50, *p* ≤ 0.05, respectively) (Table 3).

Significant correlations were observed between $V_{AT}$, $V_{RCP}$, $V_{\dot{V}O_{2}}$max and all swimming speeds: $V_{total100}$, $V_{surface}$, and $V_{STF}$. The level of ventilatory indices expressed in (L · min^−1^) did not significantly correlate with swimming results. The best predictor of swimming results was $V_{RCP}$ (Table 4).

High to very high correlations were observed between *SL* and *SI* with $V_{total100}$, but *SR* did not correlate with this speed (Table 5). The kinematic indices were not correlated with body height.

For *BA* mediation analysis we selected variables $(V_{AT},\;V_{RCP}$, $V_{\dot{V}O_{2}max}$, *SL*, and *SI*) which were significant predictors of the swimming race in this study ($V_{total100}$). This was carried out to examine the effect of maturation (*BA*), which could influence the relation of selected predictors on swimming performance.

None of the tested variables—$V_{RCP}$, $V_{\dot{V}O_{2}max}$, $V_{AT}$, *SI*, or *SL*—were identified as mediated by biological age. The mediation analysis showed moderate or strong correlations between variables and $V_{total100}$ (Figure 3a–e).

## 4. Discussion

This study showed that 100 m front crawl race results of adolescent female swimmers were significantly related to swimming endurance: $V_{AT}$, $V_{RCP}$, $V_{\dot{V}O_{2}}$ max and stroke kinematics *SL* and *SI*. The main finding is that those particular relationships were observed without significant mediation effect of *BA*. As far as we know, $V_{\dot{V}O_{2}}$ max was not presented when evaluating adolescent female swimming performance. In this study, $V_{\dot{V}O_{2}}$ max showed a significant relationship (r = 0.47, *p* ≤ 0.05) with the 100 m front crawl race, but also our choice of $V_{AT}$ and $,V_{RCP}$ as predictors of swimming performance is rare in young swimmers.

In young swimmers, even in short race distances with duration over one minute, aerobic power development is crucial [1]. Malina et al. [13] noted that aerobic power of swimmers more advanced in maturation was higher than in their less mature peers. In longitudinal studies of the development of cardiorespiratory capacity, which have rarely been conducted with young swimmers, there has been observed an ability to perform increased volume and intensity of training load [8]. Those changes linked with growth and development of aerobic power translate into an increase in swimmers′ achievements [1]. $\dot{V}O_{2\;}max$ (L ·$\;\min^{-1}$) results presented in our study show no significant correlation (r = 0.19, *p* ≤ 0.05) with 100 m race results, as in the Unnithan et al. [34] study of female adolescent swimmers (age 15.3 ± 1.5 years). Lack of significant correlations between$\dot{\;V}O_{2\;}max$ and performance in our study might be caused by not fully utilized cardiovascular and respiratory range and by use of anaerobic energy source in the 100 m race. The noted results also significantly show higher speeds of $V_{AT}$ and $V_{RCP}$ in relation to the 100 m race, which must be the advantages of better developed swimming economy and endurance of the best swimmers. In our study, the results for absolute maximal oxygen uptake (3.01 ± 0.42 L · min^−1^) were higher than in the study of Plyley, Wells and Schneiderman-Walker [35] (2.7 ± 0.3 L · min^−1^) assessed during tethered swimming. The reasons for that could be that females in our group, despite similar chronological age, were taller with greater body mass and reached greater exhaustion in the flume stage test (higher values of *RER*). The $V_{RCP}$ relationship with the 100 m front crawl race presented in our study shows a similar strength as that of critical velocity of 30 min aerobic endurance test of young females (age 11.5 ± 0.6 years) and swimming performance in their personal best events (r = 0.55) [36]. We found that a 100 m front crawl race of young female swimmers is highly correlated with an ability to maintain higher swimming intensity and endurance—$V_{RCP}$.

In this study, anaerobic tethered swimming indices were not significantly related to 100 m race but remained influenced by *BA*. Taylor S, MacLaren D, Stratton G [5] observed a substantial increase in mean force production in tethered swimming in 13-year-old swimmers, which is explained by the development of the glycolytic energy system caused by maturation. Geladas, Nassis, and Pavlicevic [37] have not found a significant relationship (r = −0.18, *p* ≤ 0.05) between grip strength and a 100 m race of young female swimmers (age 12.68 ± 0.06 years), but in male participants this correlation was strong (r = −0.73; *p* ≤ 0.01). In a study of Silva et al. [38], tethered swimming indices $F_{max}$ and $F_{ave}$ were not significant predictors of young female swimmers’ sprint performance. A study of Oliveira et al. [39] revealed the significance of controlling for the maturation effect of specific strength evaluation in adolescent swimmers, showing biological maturation mediated positively between anthropometric or body composition and the propulsive force of arms. As mentioned by Vorontsov [8], early maturers demonstrate greater physical abilities and performance level than their peers who are normal or late maturers. According to Moran et al. [6], swimmers who have already reached the peak height velocity are more likely to respond more to strength training. In our study, anthropometrics showed no significant relationship with the 100 m race, so we can state the advantage of better efficiency of swimming technique (*SI, SL*) of leaders instead of strength. However, undoubtedly, also in female swimming the appropriate level of strength must be reached. This study did not find in female swimmers significant influence of height on the 100 m race, *SL*, or *SI*, but Lätt et al. [2] observed significant correlation (r = 0.41, *p* ≤ 0.05) between *SL* and height of adolescent swimmers. Silva et al. [40] showed longer *SL* and higher height in age 11–12 female swimmers, affected by advanced calendar age and technique development of earlier maturers. Geladas et al. [37] also did not find any relationship between anthropometrics and the 100 m race in 13-year-old female swimmers, but they revealed body height, hand length, and horizontal jump association with *BA*, which explained only 17% of the variance of the 100 m race. Zuozienė and Drevinskaitė [41] reported in young female swimmers (11.8 ± 0.4) a lack of significant correlation between anthropometrics and the 200 m race. They concluded that, in girls versus boys, anthropometrics predicts swimming performance less. On the other hand, physical characteristics of young swimmers might influence swimming performance [2,11]. Toussaint and Beek [42] pointed out that young swimmers’ ability to increase maximal swimming velocity is associated with better force-generating capacity caused by age-related growth in muscle size. Vorontsov et al. [7] concluded that the strongest effect of maturity on physical development and strength could be observed in girls aged 13–14.

The relationship between stroke kinematics and the 100 m race showed that *SI* and *SL* are good swimming speed predictors, especially here for young females, because they are free of *BA* mediation. This indicates that their size depends mostly on the type of technique dedicated to swimming training. Mezzaroba and Machado [43] pointed out that *SR*, *SL*, and *SI* included in multiple linear regression of swimmers aged 10–17 could explain almost 100% (*R*^2^ = 0.99) of 100 m race results of young swimmers. Jürimäe et al. [4] stated that *SI* may be an important indicator of swimming economy in adolescent swimmers. Morais et al. [44] revealed a strong relation between adolescent girls’ (12.31 ± 1.09 years) 100 m race time result and *SI* (r = −0.82, *p* ≤ 0.01) and *SL* (r = −0.61, *p* ≤ 0.05), which is very similar to our speed performance and kinematics (r = 0.81, *p* ≤ 0.01 and r = 0.61, *p* ≤ 0.01, respectively). Lätt et al. [12] also presented high partial correlation between time of 100 m front crawl performance in young male swimmers (age 15.2 ± 1.9 years) and *SI* (r = −0.643; *p* ≤ 0.05).

Despite objective strengths of the presented study, some limitations should also be noted. For example, the method used to assess biological age is not validated, or the small sample of participants examined may limit the application of the conclusions in regards to the wide swimming community.

## 5. Conclusions

This study analyzed significant predictors of the 100 m front crawl race in adolescent female swimmers: front crawl swimming endurance ($V_{AT},V_{RCP},V_{\dot{V}O_{2}}$*max*) and stroke kinematics (*SL*, *SI*). The noted predictors were not mediated by *BA*. These results showed that young female swimmers rely on trained physiological capacity and efficient front crawl stroke technique and less on somatic traits or strength. The identified predictors are certainly susceptible to the influence of well-thought-out, planned swimming training.

### Key Points


-Biological age must be taken into consideration when evaluating young female swimmers’ abilities in regards to training and performance;-Efficiency of sprint swimming technique reflected by *SL* and *SI*, crucial in young female swimmers may be more dependent on the training used and less dependent on biological age;-Swimming speed at ventilatory thresholds and maximal oxygen uptake is valuable in terms of assessment of the physiological build-up in relation to performance in adolescent female sprint swimming.


## Figures and Tables

**Figure 1 ijerph-18-06062-f001:**
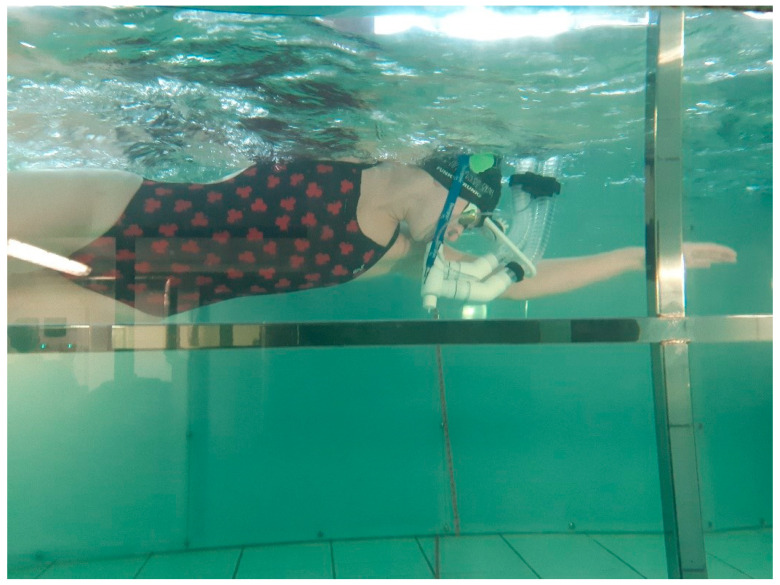
One of the swimmers going through a stage test procedure in a water flume.

**Figure 2 ijerph-18-06062-f002:**
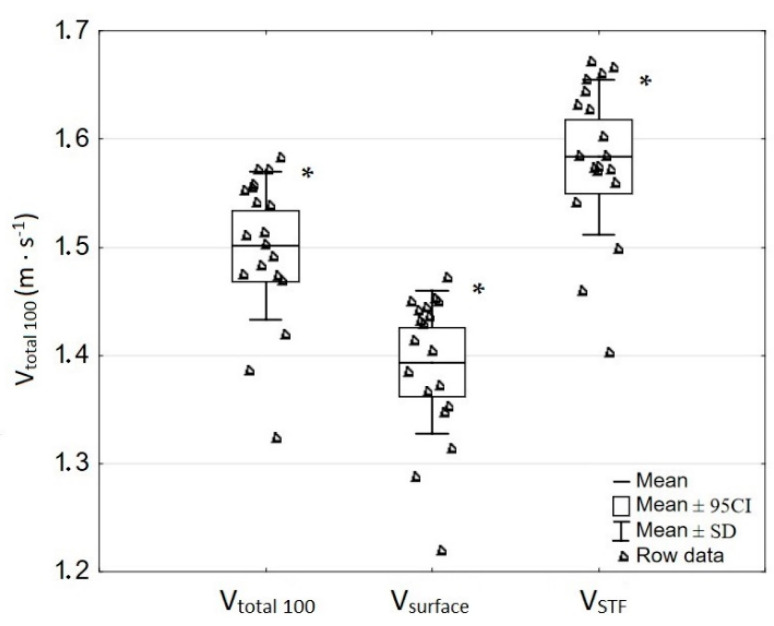
Comparison between average speed values of all of the distance ($V_{total100}$), surface swimming zones $\left(V_{surface}\right)$, and start, turn, and finish zones $\left(V_{STF}\right)$ measured during 100 m crawl stroke race. * Significant difference from the other speeds; *p* ≤ 0.001.

**Figure 3 ijerph-18-06062-f003:**
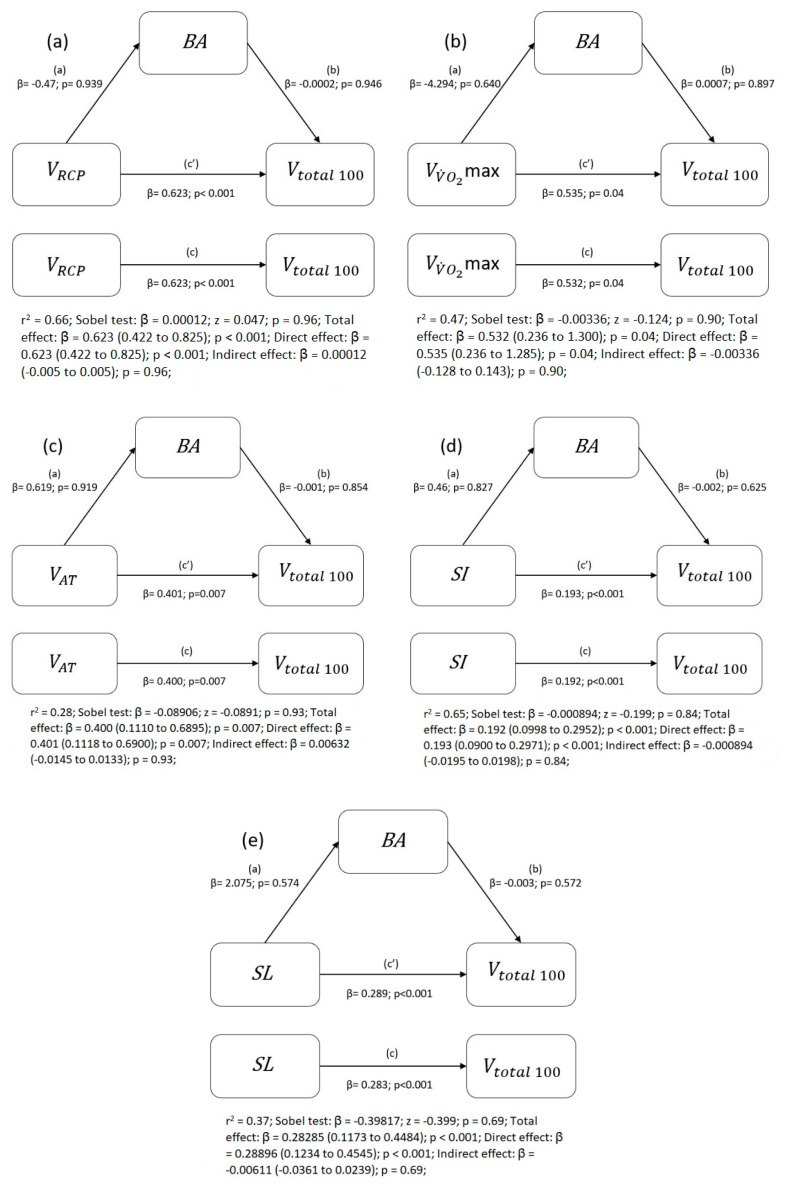
Mediation models illustrating the level of mediation effects of the relation between the independent variables of: (**a**) $V_{RCP}$, (**b**) $V_{\dot{V}O_{2}max}$, (**c**) $V_{AT}$, (**d**) *SI*, (**e**) *SL* and dependent variable of $V_{total100}$. β and corresponding *p*-values are presented. β of total (c), direct (c’), and indirect (a,b) effects are presented with 95% confidence intervals, *p*-value.

**Table 1 ijerph-18-06062-t001:** Correlations between *BA* and anthropometric, body composition indices: *h*, BM, *FFM*, *FFM*, $m_{m\;total}$. In the top row, there are mean values and standard deviations (m ± SD) of anthropometric, body composition indices with corresponding ranges (min–max) presented. In the lower line there are values of Pearson correlations with corresponding *p* values.

Correlations	h (cm) 166.0 ± 6.60 Min: 153.0 Max: 174.0	BM(kg) 55.5 ± 9.30 Min: 39.3 Max: 73.4	FFM (kg) 42.46 ± 5.54 Min: 33.8 Max: 50.1	FFM (%) 76.94 ± 3.53 Min: 70.84 Max: 83.96	TBW (kg) 31.10 ± 4.24 Min: 24.2 Max: 38.1	TBW (%) 56.35 ± 2.65 Min: 51.91 Max: 61.58	$m_{m\;total}(\mathrm{kg})$ 40.3 ± 5.50 Min: 31.3 Max: 49.5
BA (years) 15.79 ± 2.38	0.89 *p* < 0.001	0.88 *p* < 0.001	0.92 *p* < 0.001	−0.56 *p* = 0.012	0.92 *p* < 0.001	−0.56 *p* = 0.013	0.92 *p* < 0.001

**Table 2 ijerph-18-06062-t002:** Correlations of anthropometric, body composition indices: *BA*, *h*, BM, $m_{m\;total}$, $m_{m\;arms\;}$, $m_{m\;trunk}$, and $m_{m\;legs}$ with tethered swimming indices: $F_{max}$, $F_{ave}$, $I_{ave}$, and $F_{decline}$.

Correlations	BA (years)	h (cm)	*BM* (kg)	$m_{m\;total}\;(\mathrm{kg})$ 40.3 ± 5.50	$m_{m\;arms}\;(\mathrm{kg})$ 3.4 ± 0.54	$m_{m\;trunk}\;(\mathrm{kg})$ 24.0 ± 3.14	$m_{m\;legs}\;(\mathrm{kg})$ 6.4 ± 0.54
$F_{max}$ (N) 227.64 ± 46.04	0.78 *p* < 0.001	0.78 *p* < 0.001	0.63 *p* = 0.004	0.66 *p* = 0.002	0.64 *p* = 0.003	0.68 *p* = 0.001	0.61 *p* = 0.006
$F_{ave}$ (N) 79.7 ± 10.42	0.76 *p* < 0.001	0.65 *p* = 0.003	0.70 *p* = 0.001	0.74 *p* < 0.001	0.77 *p* < 0.001	0.77 *p* < 0.001	0.68 *p* = 0.001
$I_{ave\;}$(N·s) 50.6 ± 6.99	0.78 *p* < 0.001	0.71 *p* = 0.001	0.70 *p* = 0.001	0.75 *p* < 0.001	0.74 *p* < 0.001	0.77 *p* < 0.001	0.68 *p* = 0.001
$F_{decline}$ (N) 20.07 ± 8.33	0.56 *p* = 0.013	0.65 *p* = 0.003	0.53 *p* = 0.020	0.55 *p* = 0.014	0.52 *p* = 0.022	0.57 *p* = 0.010	0.51 *p* = 0.026

**Table 3 ijerph-18-06062-t003:** Correlations of stage test physiological and kinematic indices:$\;\dot{V}O_{2}AT$, $\dot{V}O_{2}RCP$, and $\dot{V}O_{2}max$ with anthropometric, body composition indices.

Correlations	BA (Years)	h (cm)	BM (kg)	FFM (kg)	TBW (kg)	F_max_ (N)	F_ave_ (N)
$\dot{V}O_{2}AT$ (L ·$\;min^{-1}$) 1.82 ± 0.09	0.52 *p* = 0.024	0.44 *p* = 0.060	0.63 *p* = 0.004	0.62 *p* = 0.004	0.62 *p* = 0.004	0.53 *p* = 0.019	0.42 *p* = 0.070
$\dot{V}O_{2}RCP$ (L ·$\;min^{-1}$) 2.47 ± 0.45	0.66 *p* = 0.002	0.63 *p* = 0.004	0.57 *p* = 0.011	0.60 *p* = 0.006	0.60 *p* = 0.006	0.50 *p* = 0.028	0.37 *p* = 0.121
$\dot{V}O_{2}max$ (L ·$\;min^{-1}$) 3.01 ± 0.42	0.45 *p* = 0.051	0.46 *p* = 0.046	0.39 *p* = 0.095	0.49 *p* = 0.032	0.49 *p* = 0.032	0.29 *p* = 0.224	0.29 *p* = 0.221

**Table 4 ijerph-18-06062-t004:** Correlations of stage test physiological and kinematics indices $V_{AT}$, $VO_{2}AT$, $V_{RCP}$, $VO_{2}RCP$, $V_{\dot{V}O_{2\;}max}$, and $\dot{V}O_{2}max$ with$\;V_{total100}$, $V_{surface}$, $V_{STF}$.

Correlations	$V_{AT}$ $(m\;\cdot \;s^{-1})\;0.92\;\pm \;0.09$	$\dot{V}O_{2}AT$ $\left(L\;\cdot \;min^{-1}\right)$	$V_{RCP}$ $(m\;\cdot \;s^{-1})\;1.15\;\pm \;0.09$	$\dot{V}O_{2}RCP$ $\left(L\;\cdot \;\min^{-1}\right)$	$V_{\dot{V}O_{2}max}$ $\left(m\;\cdot \;s^{-1}\right)$ 1.23 ± 0.06	$\dot{V}O_{2}max$ $\left(L\;\cdot \;\min^{-1}\right)$
$V_{total100}$ ($\frac{m}{s}$) 1.50 ± 0.07	0.53 *p* = 0.020	−0.10 *p* = 0.690	0.81 *p* < 0.001	0.23 *p* = 0.336	0.47 *p* = 0.044	0.27 *p* = 0.264
$V_{surface}$ ($\frac{m}{s}$) 1.39 ± 0.07	0.50 *p* = 0.031	−0.13 *p* = 0.597	0.85 *p* < 0.001	0.22 *p* = 0.357	0.55 *p* = 0.014	0.29 *p* = 0.226
$V_{STF}$ ($\frac{m}{s}$) 1.58 ± 0.07	0.54 *p* = 0.018	−0.07 *p* = 0.778	0.76 *p* < 0.001	0.23 *p* = 0.334	0.39 *p* = 0.104	0.24 *p* = 0.315

**Table 5 ijerph-18-06062-t005:** Correlations of kinematic indices of *SR*, *SL*, and *SI* with:$\;V_{total100}$, $V_{surface}$, $V_{STF}$.

Correlations	$\mathrm{SR}\;(\mathrm{cycle}\;\cdot \;\min^{-1})$ 47.31 ± 3.24	SL (m) 1.77 ± 0.15	$\mathrm{SI}\;(\frac{m^{2}}{s})$ 2.48 ± 0.29
$V_{total100}$	−0.05 *p* = 0.844	0.61 *p* < 0.001	0.81 *p* < 0.001
$V_{surface}$	0.02 *p* = 0.935	0.58 *p* = 0.010	0.79 *p* < 0.001
$V_{STF}$	−0.10 *p* = 0.681	0.62 *p* = 0.005	0.79 *p* < 0.001

## Data Availability

The data presented in this study are available on request from the corresponding author.

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
