# Peer review of "Biological Age in Relation to Somatic, Physiological, and Swimming Kinematic Indices as Predictors of 100 m Front Crawl Performance in Young Female Swimmers"

_ijerph, 2021, doi:10.3390/ijerph18116062_

Round 1

Reviewer 1 Report

The changes made by the authors are superficial, not having resolved very important issues from a methodological point of view such as the use of bioimpedance for the assessment of the population, the process for carrying out the Tanner, the method of estimating menarche, the disparity between what is indicated in the objective to be achieved and the statistical analysis carried out to achieve this objective, the editing of the tables including the means, deviations, maximum and minimum values of all the variables and the method of estimating biological age. Therefore, the article does not meet the criteria to be accepted in a journal of this category.

Author Response

Reviewer 1 Comments

The changes made by the authors are superficial, not having resolved very important issues from a methodological point of view such as the use of bioimpedance for the assessment of the population, the process for carrying out the Tanner, the method of estimating menarche, the disparity between what is indicated in the objective to be achieved and the statistical analysis carried out to achieve this objective, the editing of the tables including the means, deviations, maximum and minimum values of all the variables and the method of estimating biological age. Therefore, the article does not meet the criteria to be accepted in a journal of this category.

Response to Reviewer 1

Dear Reviewer,

Thank you for your support and commitment, and for valuable time spent improving our manuscript. We make changes in accordance with all the directions you send, we understand that not all the changes we have made are satisfactory to you. Nevertheless, the changes to the manuscript made thanks to your insight will certainly be valuable.

Sincerely

Authors

Reviewer 2 Report

Dear. Author,

Thanks for re-submission of your manuscript. 

I have pointed out in the previous peer review have generally been improved. I am not an expert in swimming research, but I believe that the methodology has been appropriately modified. 

The final judgment is left to the editors.

Best,

Author Response

Reviewer 1 Comments

Dear Author,

Thanks for re-submission of your manuscript. 

I have pointed out in the previous peer review have generally been improved. I am not an expert in swimming research, but I believe that the methodology has been appropriately modified. 

The final judgment is left to the editors.

Best,

Response to Reviewer 1

Dear Reviewer,

Thank you for your support and commitment, and for valuable time spent improving our manuscript. We make changes according to all the directions you sent.

Sincerely

Authors

Reviewer 3 Report

The authors did a good job addressing all of my issues. I am satisfied with the revisions made and have no further comments. 

Author Response

Reviewer 1 Comments

The authors did a good job addressing all of my issues. I am satisfied with the revisions made and have no further comments.

Response to Reviewer 1

Dear Reviewer,

Thank you for all your valuable comments and advice that added value to our manuscript. We've used them all.

Sincerely

Author's

Round 2

Reviewer 1 Report

You have not included information about: - The use of bioimpedance for the assessment of the population.   Furthermore, there is a challenge in the design of the study: The method of estimating menarche has not been validated.

Author Response

Response to Reviewer 1 Comments

We would like to sincerely thank the reviewers and editors for their helpful recommendations. We have seriously considered all the comments and carefully revised the manuscript accordingly during the different stages of the review. We feel that the quality of the manuscript has been significantly improved with these modifications and improvements based on the reviewers’ suggestions and comments. We hope our revision will lead to an acceptance of our manuscript for publication in International Journal of Environmental Research and Public Health.

Point 1: You have not included information about: - The use of bioimpedance for the assessment of the population.

Response 1: According to reviewer comment we added appropriate citations and explained the assessment of the BIA method more widely.

Point 2: Furthermore, there is a challenge in the design of the study: The method of estimating menarche has not been validated.

Response 2: Menarche estimating method used in our study has been validated, used in anthropological studies. We added appropriate citations of these publications just to clarify.

We would like to thank the reviewer for the effort of addressing additional valuable comments. We hope that changes made meet the reviewers’ expectations and improved design of the study. Parts of the manuscript added recently have been underlined with the green marker.

Sincerely,

Authors

Round 3

Reviewer 1 Report

The current version is enough. Thanks.

This manuscript is a resubmission of an earlier submission. The following is a list of the peer review reports and author responses from that submission.

Round 1

Reviewer 1 Report

- In the introduction, the changes that occur during puberty in the variables analysed, as well as their relationship with sports performance, are not explored in depth, focusing the argumentation on women, as they are the object of this study. Furthermore, the information within it is not interwoven nor does it have argumentative continuity.
- In the introduction you talk about the influence of anthropometric variables on swimming performance in general, but then you only measure weight, height and body composition. Please clarify exactly which parameters have been shown in previous studies to have an influence on swimming performance.
- There is a lack of information on how the selection process was used to choose the 19 swimmers from the 85 most successful swimmers.
- It would be necessary to include information about the category, competitive level, training volume, etc. of the selected swimmers.
- It is necessary to include information about the age range of the swimmers, due to its influence on the estimation of the biological age at these ages, although it is assumed to be close to the mean age given the standard deviation.
- It would be convenient to include the calculation of the sample size in the method.
- The use of bioimpedance as a method for estimating body composition in adolescent females, especially if they are involved in sport, has some limitations. It would be appreciated to include in the method a study that has demonstrated the validity of this technique in this population, since the citations included in the article are in totally different populations. 
- Furthermore, bioimpedance is not only used to obtain general body values, but also to distinguish by limb. Please justify your choice based on previous work.
- Please explain the abbreviations BHage and BMage the first time you use them.
- Could you please include citations supporting the use of the method explained for the calculation of biological age in this population, as opposed to others rather more frequently used in previous studies?
- How were the Tanner stages assessed, by self-report or by external assessor?
- The method for obtaining the date of menarche is highly imprecise. Please justify your choice based on previous work.
- Please clarify why you do a Kolmogorov-Smirnov normality test when your sample is n=19.
- There is disparity between what is stated in the objective you intend to do and the statistical analysis you perform to achieve this objective. Please readjust this.
- In order to see the characteristics of the sample in the variables analysed, a first table showing the means, deviations, maximum and minimum values of all the variables would be appreciated.
- Could the lack of association of biological age with some performance parameters be due to the method chosen for the estimation of biological age? This should be discussed as it is an unvalidated method.

Reviewer 2 Report

Dear. author,

The experimental procedure of this study was done in a standard way in swimming research and I do not see any problems. However, I would like to ask two questions. 

(1) You have undergone an ethical review, but the participants are under 20 years of age. There is a description that parental consent was obtained, but it is necessary to describe in detail what procedures were used.

(2) Regarding the participants, the details of the swimming level are not described. Also, Crawl performance is verified, but are all of the 19 participants' specialties freestyle? I think that adding such information would make this study more interesting.

Best,

Reviewer 3 Report

The aim of the current study was to analyse the influence of a set of morphological, strength, and physiological indices on the 100-m front crawl swimming performance in adolescent female swimmers. Additionally, the authors sought to identify a set of variables which influence 100-m front crawl performance in female swimmers, but which are not directly related to biological age. This is a well-prepared manuscript. I have only minor comments: Abstract Line 19: Remove “of the” Line 20: Change to “The purpose of …” Line 23: replace the comma with “and”. Line 23-24: Issues with grammar here, rephrase. Line 27: Issues with grammar here, rephrase. Can you include correlation values in the abstract, even ranges if there are too many. Introduction You have many short paragraphs, some of them can be combined. Line 60: Higher by how much? Methods Line 106-107: Please specify who assessed the Tanner stages. Line 121-122: Provide a justification for these speeds. Line 123-126: I believe there is some punctuation missing here. Also, revise whether the use of capitals is necessary. Line 165: Why was the Kolmogorov-Smirnov test used? Line 168: How were the strengths of the correlations interpreted? Include information about the study’s sensitivity based on you sample size of 19 and your chosen power. Results Figure 2: Can you include a symbol to denote significant differences. Table 1 & Table 2: Replace “p = 0.000” with “p < .001”. Discussion Line 243: Remove “the” Good discussion in relation to the extant literature.